# The Longitudinal Analysis of Convergent Antibody VDJ Regions in SARS-CoV-2-Positive Patients Using RNA-Seq

**DOI:** 10.3390/v15061253

**Published:** 2023-05-26

**Authors:** Kate J. Liu, Monika A. Zelazowska, Kevin M. McBride

**Affiliations:** Department of Epigenetics and Molecular Carcinogenesis, The University of Texas MD Anderson Cancer Center, Houston, TX 77030, USA; kliu4@mdanderson.org (K.J.L.); mazelazowska@mdanderson.org (M.A.Z.)

**Keywords:** SARS-CoV-2, complementarity-determining region 3, convergent, RNA-seq

## Abstract

Severe acute respiratory syndrome-related coronavirus-2 (SARS-CoV-2) is an ongoing pandemic that continues to evolve and reinfect individuals. To understand the convergent antibody responses that evolved over the course of the pandemic, we evaluated the immunoglobulin repertoire of individuals infected by different SARS-CoV-2 variants for similarity between patients. We utilized four public RNA-seq data sets collected between March 2020 and March 2022 from the Gene Expression Omnibus (GEO) in our longitudinal analysis. This covered individuals infected with Alpha and Omicron variants. In total, from 269 SARS-CoV-2-positive patients and 26 negative patients, 629,133 immunoglobulin heavy-chain variable region V(D)J sequences were reconstructed from sequencing data. We grouped samples based on the SARS-CoV-2 variant type and/or the time they were collected from patients. Our comparison of patients within each SARS-CoV-2-positive group found 1011 common V(D)Js (same V gene, J gene and CDR3 amino acid sequence) shared by more than one patient and no common V(D)Js in the noninfected group. Taking convergence into account, we clustered based on similar CDR3 sequence and identified 129 convergent clusters from the SARS-CoV-2-positive groups. Within the top 15 clusters, 4 contain known anti-SARS-CoV-2 immunoglobulin sequences with 1 cluster confirmed to cross-neutralize variants from Alpha to Omicron. In our analysis of longitudinal groups that include Alpha and Omicron variants, we find that 2.7% of the common CDR3s found within groups were also present in more than one group. Our analysis reveals common and convergent antibodies, which include anti-SARS-CoV-2 antibodies, in patient groups over various stages of the pandemic.

## 1. Introduction

When humans are infected with severe acute respiratory syndrome-related coronavirus-2 (SARS-CoV-2), the immune system will generate antibodies with specificity to SARS-CoV-2 and these antibodies can be detected in the blood serum of patients [1,2,3]. The antibodies that neutralize SARS-CoV-2 play an important role in mitigating the severity of the coronavirus disease-2019 (COVID-19) [4,5]. The characterization of antibodies helps elucidate the mechanisms of immune response and guide strategies for the potential treatment of COVID-19. Immunoglobulins (Ig) with highly diversified antigen binding specificity are generated by B-lymphocyte development during V(D)J rearrangement of their immunoglobulin (Ig) genes [6]. The Igs are initially expressed as B cell receptors (BCRs) on the cell surface, which enable B cell response to antigens. The variable domain of heavy and light chains contains complementarity-determining regions (CDRs), loops at the antibody–antigen surface, which are determinants of affinity and specificity. The CDR3 of the heavy chain covers the D gene and junction regions of D-J and V-D. The highly variable sequence of the CDR3 region is a major contributor of antibody diversity, which is theoretically estimated to be above 10^15^ variants [7,8]. During an immune response, B cells that express BCRs with affinity to an antigen can be activated to undergo affinity maturation during the germinal center reaction [3]. Through a combination of somatic hypermutation and selection, BCRs with high affinity are evolved and clonally expanded. Out of this process, antibodies with high specificity and affinity will develop. The antibody response to subsequent SARS-CoV-2 variants may derive from ones developed against previous infections or be newly generated. 

Thousands of anti-SARS-CoV-2 antibodies have been cloned and characterized from COVID-19 patients. Certain antibodies cloned from patients infected with an early SARS-CoV-2 variant (WA-1) have been reported to be able to interact with a wide range of variants of concern (VOCs) [9]. Antibodies against different VOCs with broad neutralization have been identified [10,11,12,13], with many of these being public antibodies [4,14,15]. Public or common antibodies are described as antibodies arising from different donors but with the same genetic elements (IGHV) and CDR3 amino acid sequences, which result in comparable antigen recognition. Within an individual, B cells with a common immunoglobulin sequence are described as clonally derived; however, among individuals, these common/public antibodies are the result of convergent evolution. Public antibodies reactive against SARS-CoV-2 suggest that infected subjects select antibodies with a common structural basis to target specific epitopes [4,14]. As different SARS-CoV-2 variants are constantly evolving and appearing worldwide, it is useful to investigate convergent antibodies that arise and maintain presence across different time periods of the COVID-19 pandemic.

Bulk RNA-seq is a widely used methodology for the differential expression (DE) of genes [16,17,18]. Routine DE analysis of RNA-seq mainly focuses on gene expression level and is generally not suitable for focused immunoglobulin sequence analysis and comparison. Although single-cell or direct BCR-seq sequencing facilitate direct V(D)J sequencing [8,19], bulk RNA-seq data sets are much more widely available. However, RNA-seq of lymphocyte transcripts using high-throughput next-generation sequencing does provide data that cover expressed immunoglobulin genes. Therefore, we utilized immune repertoire reconstruction tools to retrieve V(D)J sequences from Bulk RNA-seq [20,21]. In this study, we leveraged the TRUST4 to reconstruct the heavy-chain VDJ for a longitudinal analysis of convergent antibodies from sequenced COVID-19 patient samples.

## 2. Materials and Methods

**RNA-seq data sets**: Four public RNA-seq data sets were retrieved from the Gene Expression Omnibus (GEO) [22]. The fastq-dump from the SRA-Toolkit (https://hpc.nih.gov/apps/sratoolkit.html, accessed on 29 July 2022) was used to download the RNA-seq data sets. Collectively, there were 269 COVID-19-positive patients and 26 negative patients. These patients were distributed among 5 different groups based on the variant they were infected with (Figure 1a). COVID-19-positive and -negative patients in Group 2 were split into two subgroups (2A and 2B). Groups 3 and 4 were from the same data set but consisted of patients infected with different COVID-19 mutations. The data sets in this study satisfied the following criteria: (1) samples were collected from patient’s blood PBMCs; (2) RNA-seq data had a high sequence depth with a minimum of 10 million reads per sample (if a sample had multiple read files, then they were combined into one read file); (3) each sample had either a clear variant identification or date when the sample was collected in order to define the predominant variant.

Samples in Group 1 were collected between March 2020 and April 2020. The data set consisted of 69 blood samples which were extracted from patients who were admitted to either the infectious disease unit or the designated ICU at a university hospital network in northeast France. All the patients in this group tested positive for SARS-CoV-2 through a qRT-PCR test which detected COVID-19 nucleic acids using a nasopharyngeal swab [23].

Samples in Group 2 were collected between April 2020 and May 2020 from patients that were admitted to either Albany Medical Center’s medical floor or the medical intensive care unit (MICU). The data set included 126 blood samples from 100 patients who tested positive for SARS-CoV-2 and 26 patients with varying respiratory issues who tested negative for SARS-CoV-2 [22].

Samples in Group 3 were collected between February 2021 and May 2021 by extracting blood from the buffy coat which was then purified with the Maxwell RSC simply RNA Blood Kit. In total, the data set included 100 blood samples which were extracted from 48 patients. We assigned 32 of the 48 patients to Group 3 as they had been classified as patients who had contracted the Alpha variant. Blood was extracted from patients through three samplings which took place 10 days, 25 days and 45 days after the patient began feeling symptoms compatible to SARS-CoV-2. All 32 patients had their blood sampled 10 days after the patient started experiencing symptoms, 30 samples were collected 25 days after the patient started experiencing symptoms and 5 samples were collected 45 days after the patient started experiencing symptoms. In our experiment, we combined each patient’s samplings into one sample [24,25].

Samples in Group 4 were from the same downloaded data set as Group 3 but were classified as patients who had contracted the E484K escape mutation of the Alpha variant. There were 13 patients who tested positive for the mutation and samples were collected through three different samplings. All 13 patients were sampled 10 days after experiencing symptoms, 10 patients were sampled 25 days after experiencing symptoms and 7 patients were sampled 45 days after experiencing symptoms. In this group, each patient’s samplings were combined as well [24,25].

Samples in Group 5 were collected between December 2021 and March 2022 from patients that were infected with the Omicron variant. Patients in this group were admitted due to either testing positive after attending an outpatient clinic at Krankenhaus St. Vinzenz Zams, Austria, using the SARS-CoV-2 PCR test or being in contact with a family member who had tested positive for SARS-CoV-2. In total, 47 of the blood samples were collected zero, one, two, three, four and five days after patients had tested positive for SARS-CoV-2 by the qRT-PCR test. Overall, 14 patients were sampled on day 0, 7 patients were sampled on day 1, 14 patients were sampled on day 2, 7 patients were sampled on day 3, 2 patients were sampled on day 4 and 3 patients were sampled on day 5 [24].

**Reconstruction of VDJs and Longitudinal Analysis**: The workflow is shown in Figure 1b. The RNA-seq raw reads from each single patient was merged into one fastq file and used as input for the bioinformatics tool TRUST4 [20,26]. TRUST4 reconstructs immune repertoires and annotates the V(D)J assembly of each sample. Then, using the reconstructed immunoglobulin molecules, the variable regions, including the frameworks and hypervariable regions, are annotated. The annotated AIRR formatted files were sorted in each group by their V gene and J gene usage and their CDR3 sequence. Common VDJs are defined when two or more patients in a group have the same V gene and J gene usage and the same amino acid CDR3 sequence. This definition matches that of previous research studies [27,28].

The normalized scores to evaluate the frequencies of common VDJs were calculated. Each group consisted of a different number of patients and a different number of VDJs predicted in each patient; therefore, the normalized score included the weight of the number of patients in each group. Thus, we could perform longitudinal comparison without the effects of each group’s population size. To calculate the normalized score of common VDJs, we implemented the following equation:$$\mathrm{Normalized}\;\mathrm{Score}\;\mathrm{for}\;\mathrm{Common}\;V\left(D\right)\mathrm{Js}=\frac{n_{x}}{N_{x}}\cdot 100\cdot \frac{S}{s_{i}}$$

*n_x_* = number of common V(D)Js in patient *x**N_x_* = total number of V(D)Js in patient *x**S* = total patients in all groups*s_i_* = number of patients in group *i* (*i* = 1, 2, 3, 4, 5)

Normalized scores for each patient are shown as violin plots in Figure 2b. The heavy-chain IGHV gene usage was evaluated by the abundance of each IGHV gene in the common VDJs. A heatmap was created to display the frequency of each IGHV gene in Figure 2c.

**Identification of convergent VDJ clusters and Protein Data Bank search with the reconstructed VDJ sequences**: We performed a pair-wise calculation using the Levenshtein distance [29] between CDR3 amino acid sequences from groups with the same IGHV and IGHJ genes and the same amino acid sequence length of CDR3s. We set the Levenshtein distance cutoff value to 2. The network was then visualized with the Fruchterman Reingold layout from the Gephi Software (downloaded from https://gephi.org on 5 October 2022).

The common VDJ sequences were then used as queries to search the Protein Data Bank (PDB) to find homologous VDJs from an antibody that was clinically proven to be against SARS-CoV-2. We then viewed a 3D crystal structure of the antibody using the RCSB PDB Mol* Viewer 2.5.8.

## 3. Results

### 3.1. Identification of Common V(D)Js from RNA-Seq and the Longitudinal Analysis

We investigated the immunoglobulin repertoire of COVID-19 patients across different time points of the pandemic, infected with different variants of SARS-CoV-2. The goal was to identify common immunoglobulin heavy-chain sequences (common VDJs) that were found in multiple COVID-19-positive patients at each stage of the pandemic. We analyzed sequencing data sets from PBMCs that were extracted from four public RNA-seq data sets from the Gene Expression Omnibus (GEO) [22]. These were divided among five different groups based on the based on their SARS-CoV-2 variant type and/or the time they were collected (Figure 1a). Altogether, there were 269 COVID-19-positive patients and 26 negative patients. COVID-19-positive and -negative patients in Group 2 were split into two subgroups (2A and 2B). Groups 3 and 4 were from the same data set but consisted of patients infected with different COVID-19 mutations. We reconstructed the V(D)Js using the immune repertoire reconstruction tool TRUST4 [20] in our workflow (Figure 1b). A total of 629,133 heavy-chain VDJs from all patients were reconstructed and analyzed (Appendix A). The IGHV, IGHJ and CDR3 amino acid sequences were annotated for each sequence based on the VDJ genes from the international ImMunoGeneTics information system (IMGT) [30,31] and the human genome reference (version hg38). We then compared the VDJ sequences between patients within a group to find common VDJs. A common VDJ sequence is an immunoglobulin sequence utilizing the same IGHV and IGHJ genes, having identical CDR3 amino acid sequences and appearing in two or more patients. Our analysis first centered on the occurrence and number of common VDJ sequences that occurred within each separate group. We identified within Group 1 328 common VDJs; Group 2A had 125 common VDJs; Group 3 had 406 common VDJs; Group 4 had 56 common VDJs; and Group 5 had 96 common VDJs (Figure 2a, red bars). We found that within each group, a common VDJ occurred in 2 to 22 patients. The COVID-19-negative group (Group 2B) had no common VDJs within the group. 

In order to determine if the common VDJ sequences found within each group were present as common VDJ sequences in other groups, we conducted a longitudinal comparison. Each common VDJ sequence was cross-checked to other groups to see if it occurred as an Inter-Group Common VDJ (IGC-VDJs). Most were found as a common VDJs only within their own group (Figure 2a, black bars); however, 95 were IGC-VDJs (Figure 2a, blue circles connected by lines), with the number of IGC-VDJs having that pattern of appearance indicated (Figure 2a, black bars). One such IGC-VDJ (IGHV4-59, IGJH4 and CDR3 sequence ARGFDY) was observed in all five sample groups (Figure 2a). Five other VDJ sequences were found in four out of the five groups. When comparing Group 3 (infected by Alpha variant) to Group 5 (infected by Omicron variant), we found 13 common VDJs that existed in both groups, making up 2.7% of the common VDJ sequences in the two groups.

In order to compare the occurrence of common VDJs longitudinally (Inter-Group Common VDJs as IGC-VDJs), the frequencies of IGC-VDJs in common VDJs of each group were adjusted by the number of patients in each group (see the calculation of normalized score in Materials and Methods). The result shows a significant decrease in the normalized score for the patient group infected with the Omicron variant (Figure 2b). The normalized score for the SARS-CoV-2-negative groups was zero, further suggesting that the immunoglobulins with common VDJs found in other groups correlated with the viral infection.

To characterize whether common VDJ sequences had a structural relationship, we generated a heatmap of the IGHV gene usage of each group’s common VDJ sequences. The comparison of relative frequency in all five groups revealed a bias toward the IGHV3 gene family in all groups. However, there is a heavier bias toward IGHV3-30 in earlier COVID-19 pandemic groups (Groups 2 and 3) infected by the Alpha variant. Omicron-infected Group 5, from later in the pandemic, had a lower frequency of IGHV3-30 with a heightened bias towards IGHV3-74 and IGHV3-7 in its common VDJ sequences.

We downloaded the CoV-AbDab_210323.csv file (https://opig.stats.ox.ac.uk/webapps/covabdab/static/downloads/CoV-AbDab_210323.csv, accessed on 22 May 2023) from the CoV-AbDab which contains a total of 12,021 antibody records [32]. Using the 95 intergroup common VDJs (IGC-VDJs) from our study, we found that 17 of the 95 IGC-VDJs matched with heavy chains of SARS-CoV-2-binding antibodies from the CoV-AbDab records. Matching antibodies in the database contained the same IGHV, IGHJ and CDR3 regions as the IGC-VDJs from our study. The records also suggest that seven of the matching IGC-VDJs bind to several SARS-CoV-2 variants (Alpha, Beta, Delta, Gamma, Epsilon and Omicron), and nine have SARS-CoV-2-neutralizing characteristics. Details regarding the matching IGC-VDJs can be found in Appendix A.

### 3.2. Identification and Verification of Convergent VDJs

In general, the heavy-chain variable domain and CDR3 composition are the key antigen specificity determinants [33]. Immunoglobulins with the same V gene and J gene and similar CDR3 sequences have been suggested to recognize the same antigens [4,14,34]. Therefore, we combined the CDR3 length and Levenshtein distance of amino acid sequences to characterize the CDR3 regions. We classified Ig heavy chains with the same V gene, J gene and same CDR3 length with a Levenshtein distance of 2 or less as a convergent cluster.

The 1011 common VDJ sequences were plotted as nodes in a network visualization (Figure 3a). We identified a total of 129 convergent clusters (Appendix A which were noted by interconnection (Figure 3a, blue lines). The number of nodes in a cluster was scaled by color and the top 15 clusters with the most nodes are labeled (Table 1, Figure 3a). Clusters 2 and 15 covered all five SARS-CoV-2-positive groups (Table 1). We observed that Clusters 1, 2, 8 and 15 covered the Omicron-infected group and Alpha-infected groups. Clusters 3–7 and 9–14 were from groups that were not Omicron-infected (Table 1).

With our noted pattern of common sequences in COVID-19 patients, we sought evidence that they were related to response against the virus. We surveyed whether convergent clusters had reported anti-SARS-CoV-2 specificity by searching the published literature. Thousands of antibodies against SARS-CoV-2 have been sequenced and their binding characterized. A recent systemic survey of these has revealed clusters of IGHVs and convergent CDR3 sequences associated with public SARS-CoV-2 antibodies [35]. We compared our top 15 convergent clusters to the described public antibodies and convergent CDR3 sequences. From our analysis, 4 of our top 15 clusters (Clusters 2, 8, 9 and 15) have convergent or exactly matching CDR3s and IGHVs to antibodies with neutralizing or binding capacity against SARS-CoV-2 [15]. A search of the Protein Data Bank (PDB) revealed the structures of two F_ab_ structures with anti-SARS-CoV-2 specificity, that had highly similar or the exact same CDR3 amino acid sequences as members of our identified convergent Cluster 2 (Figure 3b,c) [36,37]. The Fab fragment C102 is from a neutralizing antibody against the ACE2 receptor-binding domain (RBD) of SARS-CoV-2 [36]. P5A-3A1 is from another neutralizing antibody that shows bivalent binding and inhibition of SARS-CoV-2 [37]. The complete list of 129 convergent VDJs is available in Appendix A. 

Our analysis of IGC-VDJ sequences only included VDJ sequences that were common within groups. We determined if including VDJ sequences (same IGHV, IGHJ and exact CDR3 amino acid sequence) that appeared more than once in all data sets would include more confirmed SARS-CoV-2 antibodies. The number of shared common VDJ sequences was greater (Appendix A), 1337 total. In this case, nine VDJ sequences were shared among all COVID-19-positive groups. However, the vast majority of VDJ sequences were only observed a single time (Appendix A). Comparing these nine to characterized SARS-CoV-2-binding antibodies, we find that two of those groups had identical IGHVs and similar CDR3 sequences to the SARS-CoV-2-binding antibodies in our cluster 2 (Table 1), but different IGHJs. 

## 4. Discussion

Defined as antibodies with genetic and sequence similarity in unrelated individuals, common or public antibodies have been observed in different infectious diseases. Several studies have reported common immunoglobulin sequences from infected COVID-19 patient cohorts [4,35,38]; however, these were surveyed at distinct time periods. The extent to which these common SARS-CoV-2 antibodies persist as the virus continues to evolve and reinfect individuals is not clear. In this study, we used a bioinformatic pipeline and RNA-seq data deposited in the GEO database to identify common immunoglobulin heavy-chain VDJ sequences in COVID-19 patients. We first identified common VDJ sequences found in each patient set (Figure 1a). We then compared each common VDJ sequence across patient sets. These featured patients with early variants (March/April 2020) through to the latest analyzed set, when the Omicron variant was predominant (December 2021–March 2022). We then interrogated whether common VDJ sequences were found across patient sets. There was only one VDJ sequence that was a common sequence in all the COVID-19 patient sets examined (Figure 2a). This VDJ sequence, IGHV4-59, IGHJ4 (ARGFDY) [39,40], has been reported in antibodies that recognize the S2 domain of the SARS-CoV-2 S2 domain of the spike protein. Notably, this VDJ sequence was not a common antibody in the SARS-CoV-2-negative set (Group 2b) and has been previously reported in public antibody sequences [35]. Our analysis identified five other VDJ sequences that were a common sequence in four of the five patient sets, but only three of these were present in both the earliest (Group 1) and latest (Group 5) time periods. Two of these, ARDYGDFYFDY [14,41] and ARGYGDYYFDY [14,41], have also been identified as VDJ sequences in antibodies that bind the SARS-CoV-2 RBD domain. 

In total, there were 95 IGC-VDJ sequences, with none in the COVID-19-negative group (Group 2b). Most were among Groups 1–4 and Group 5 had the fewest with 15 IGC-VDJs. Therefore, it is notable that 3 of these 15 belong to previously authenticated SARS-CoV-2-reactive antibodies. Group 5 was collected during the period when the Omicron variant predominated. It had the lowest normalization score (Figure 2b) and the IGHV usage was the most diverse compared to the other groups (Figure 2c). The immune escape and drift of SARS-CoV-2 may account for this shift. However, a change in immune exposure due to variables such as masking and social distancing could also contribute to this. Nonetheless, these three VDJ sequences were present as a common antibody across the groups, suggesting that these VDJ sequences continue to be engaged. 

We noted that some common sequences had the same V, J and very similar CDR3, so we analyzed convergence among the common sequences. Based on similar CDR3 amino acid sequences, we identified 129 convergent clusters and 15 major nodes (Figure 3, Table 1). Most of the convergent VDJ clusters are from multiple groups, which indicates that different SARS-CoV-2 VOCs occurring in different pandemic periods may induce a similar clonality of B-lymphocytes [42]. This phenomenon has also been observed in the influenza virus vaccination [43]. Comparing our top 15 to analyzed, authenticated anti-SARS-CoV-2 antibodies, we find that 4 nodes are VDJ sequences in these antibodies. Cluster 2 was found in broad neutralizing antibodies against the RBD domain of Alpha and later variants [9]. The structures of these antibodies have been determined and deposited in the PDB database (Figure 3b,c). Our Cluster 8, 9 and 15 VDJ sequences match the convergent clusters of sequences from the public antibodies analysis in [15]. Our Cluster 8 matches their Cluster 6 [15], our Cluster 9 matches their largest cluster (Cluster 1) [15] and our Cluster 15 closely matches their Cluster 10. Analysis beyond our top 15 reveal other clusters, such as Cluster 24, which is also a public anti-SARS-CoV-2 antibody (Cluster 10) [15].

Since the outbreak of the COVID-19 pandemic, a huge investment has been made to identify antibodies that have neutralizing capacity. Methods including single-cell BCR sequences and screening libraries of patient B cells and expressed antibodies from their PBMCs are labor-intensive and costly. As SARS-CoV-2 continues to evolve immune escape mutations, it is valuable to understand how patient BCRs are evolving and what antibodies may continue to have neutralizing effects. In this study, we mined RNA-seq data, which are much more widely available than BCR or scRNA-seq data, to analyze the convergent antibody response in patients. This method identified many VDJ sequences from authenticated neutralizing antibodies. Of interest is the common and convergent VDJ sequences that were prominent between patient groups. Our method does not permit us to validate against SARS-CoV-2 since we do not have matching Ig light chains. However, cross-checking scRNA-seq data accumulated from patients might allow such matching. Notably, our method identified VDJ sequences found in antibodies with cross-variant neutralizing activity. The continued analysis of patients infected with emerging VOCs may also reveal antibodies with broad neutralizing activity. 

In our study, we used BCRs derived from deidentified patient PBMCs from publicly deposited data. A limitation of this study is that we were unable to correlate patient history or pathology to the public antibodies’ presence or profile in depth. Another limitation is that we only analyzed Ig heavy-chain sequences and the connection to authentic SARS-CoV-2-recognizing antibodies relied on published sequences. As we did not have matching light-chain sequences, we did not conduct an antibody binding analysis. While a number of the IGC-VDJ sequences are found in SARS-CoV-2-recognizing antibodies, most of the public VDJ sequences identified in this study do not have associated antibody binding data. We speculate that some of these the IGC-VDJ sequences are from anti-SARS-CoV-2 antibodies and could potentially serve as candidates for future investigation. 

As SARS-CoV-2 continues to evolve and infect patients, it is informative to track convergent antibody response longitudinally. The methodology used in this study, with bulk RNA-seq, allows deep BCR profiling and sequence comparison that identify public VDJ sequences. In the future, this could be utilized as a measure of antibody response longevity. As single-cell data continue to be collected, IGC-VDJ from bulk RNA-seq analysis could be used to identify and prioritize antibodies for binding validation against emerging SARS-CoV-2 variants of concern.

## Figures and Tables

**Figure 1 viruses-15-01253-f001:**
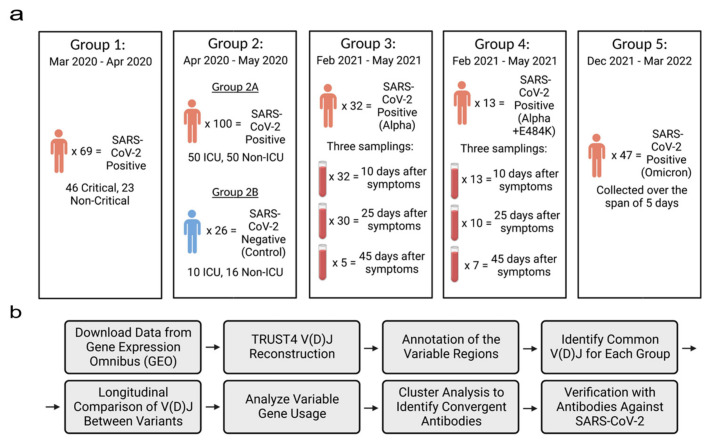
Overview of sample population and experimental design. (**a**) RNA-seq data from the same patient were combined in each group. All data were downloaded from the GEO databases (Group 1: GSE172114; Group 2: GSE157103; Groups 3 and 4: GSE190680; and Group 5: GSE201530). SARS-CoV-2 variants in Groups 1 and 2 were not identified. (**b**) The workflow of the longitudinal analysis of reconstructed V(D)Js. Figures were created with the web-based application BioRender, (https://app.biorender.com, accessed on 12 October 2022).

**Figure 2 viruses-15-01253-f002:**
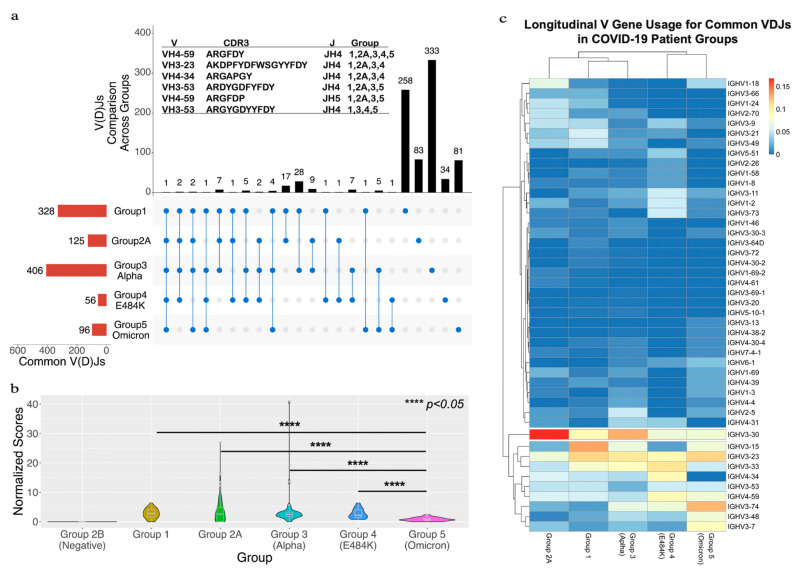
A longitudinal comparison of common VDJs across the five different groups. (**a**) The red bars at bottom left side show the total number of common VDJs for each group. The black bars at top show the number of IGC-VDJs and the number of common VDJs in only each group itself. The connected blue dots indicate the common VDJs that are shared across groups. The individual blue dots show common VDJs within one group. The panel (**a**) also lists the V, CDR3 and J for the IGC-VDJs that were found in 4 or 5 groups. (**b**) Shows a comparison of common VDJs with the normalized scores. **** represents the *p*-value < 0.05 threshold using the one-tail *t*-test with heteroscedastic setting. (**c**) The heatmap of IGHV usage in common VDJ sequences between groups. The relative frequencies each IGHV gene that occurred for the indicated group is depicted in the heatmap. Individual IGHV genes displaying a bias were grouped in the lower part of the heatmap. Lines indicate clusters of IGHVs that have similar usage.

**Figure 3 viruses-15-01253-f003:**
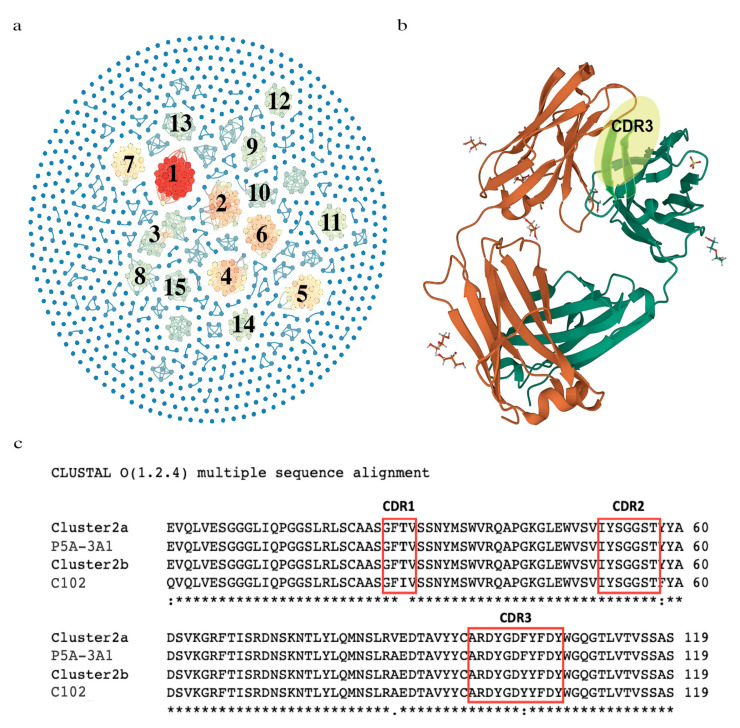
The identification of convergent VDJ clusters and comparison of the variable regions between the convergent cluster and the antibodies responsive to SARS-CoV-2. (**a**) Network visualization of convergent clusters. Related clusters are connected by blue lines and colored based on the number of nodes in each cluster. (**b**) A crystal structure of an anti-SARS-CoV-2 human neutralizing antibody Fab fragment (C102). The heavy-chain CDR3 region is highlighted in yellow. (**c**) VDJ amino acid sequence comparison among two members of the convergent Cluster 5, C102 (PDB code: 7K8N_A) and P5A-3A1 (PDB code: 7D0C_G) from an antibody recognizing SARS-CoV-2 in the Protein Data Bank (PDB). Notation Key: (*) indicates the conserved residue, (:) indicates conservation between amino acid groups of strongly similar properties, (.) indicates conservation between amino acid groups of weakly similar properties; as defined by Cluster O application.

**Table 1 viruses-15-01253-t001:** Representative CDR3 sequences, V genes and J genes for the top 15 convergent VDJ clusters.

Convergent Cluster ID	IGHV Gene	Representative CDR3	JH Gene	Group	Unique VDJs in Clusters
1	IGHV3-15	DYYHYYGMDV	IGHJ6	1, 3, 5	18
2	IGHV3-53	ARDYGDYYFDY	IGHJ4	1, 2A, 3, 4, 5	12
3	IGHV3-15	TTGGAV	IGHJ4	1, 2A, 3	15
4	IGHV3-33	ARVASYYYGMDV	IGHJ6	1, 2A, 3	11
5	IGHV3-33	AREGIVGATTGLDY	IGHJ4	1, 3, 4	10
6	IGHV4-34	ARGAPGF	IGHJ4	1, 2A, 3, 4	8
7	IGHV3-49	TRHDFWSGYYFDY	IGHJ4	1, 2A, 3	11
8	IGHV3-30	ARARGGSYYYGMDV	IGHJ6	1, 3, 5	12
9	IGHV3-53	ARDLVVYGMDV	IGHJ6	1, 2A, 3	8
10	IGHV3-30	ARDGGSWFDP	IGHJ5	1, 2A, 3	9
11	IGHV3-49	TRDDFWSGYYDY	IGHJ4	1, 2A, 3	8
12	IGHV3-23	AKDPFYDFWSGYYYDY	IGHJ4	1, 2A, 3, 4	5
13	IGHV4-31	ARVRITMIVVVDAFDI	IGHJ3	1, 2A, 3, 4	7
14	IGHV3-23	AKDRGNDYGDQLDY	IGHJ4	1, 2A, 3	8
15	IGHV4-59	ARGFDF	IGHJ4	1, 2A, 3, 4, 5	4

## Data Availability

The GEO accession numbers used in this study are Group 1: GSE157103; Group 2: GSE172114; Group 3 and 4: GSE190680; and Group 5: GSE201530.

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
