# Peer review of "The Longitudinal Analysis of Convergent Antibody VDJ Regions in SARS-CoV-2-Positive Patients Using RNA-Seq"

_viruses, 2023, doi:10.3390/v15061253_

Round 1

Reviewer 1 Report

Comments and Suggestions for Authors

The authors analyzed the VDJ regions of immunoglobin heavy chain for the RNAseq data deposited from the Gene Expression Omnibus (GEO). The study might be useful for the colleagues in the field as SARS-CoV-2 continues to circulate in the community. The manuscript was well-written, the results presented were clear and concise, the figures were nice to illustrate the complicate datasets. I only have some minor comments so that the quality of the manuscript should be improved.

As far as I understand, the authors did not perform any wet lab experiments. All of the data was based on the others’ submission to the GEO. The authors should list out the search strategy (the date of the search conducted), eligibility criteria (criteria to include and exclude), study selection (no. of colleagues performing this), data extraction for the dataset included in your analysis.

The limitations of the analysis should be discussed near the end of the manuscript.

Reviewer 2 Report

Comments and Suggestions for Authors

This paper utilizes RNA-seq data from public database to analyze the antibody VDJ sequence in COVID-19 infected patient samples, from which useful information about common COVID-specific antibody can be obtained. No wetlab is involved.

Issues in this paper:

Please use this website https://opig.stats.ox.ac.uk/webapps/covabdab/ to search / compare SARS-CoV-2 specific antibodies.

The method you used in this paper (bulk RNA seq) does not prove sufficient for identifying COVID-specific antibody response, as antigen-specific B cells only account for 0.1~1% of the total B cells in infected population, and it will continue to drop the longer after recovery. The small number of your negative control group (2B) may be hard for you to exclude the non-specific genes as 99% of the VDJ sequences are not COVID-specific. There is no direct evidence showing the VDJ+CDR3 genes you identified are COVID specific, as no light chain gene information pairing the heavy chain is known, although most of the VH genes you identified correspond to known neutralizing antibodies isolated.  There is few information about the antigen each heavy gene can bind. (although it is well known that most antibody response in COVID infection targets Spike protein, especially RBD region.)

For the common VDJ antibody (IGHV4-59, IGHJ4 (ARGFDY)) reported in this paper, I found 16 entries on the website above. Non of them are neutralizing antibodies, but they do bind to SARS-1 and SARS-2, indicating it is a cross-reactive antibody sequence. Please add the citation of these reported antibodies. 

It is worth analyzing (or at least mentioning) the light chain VJ sequence as well, and see if you can identify any commonly used light chain /CDR3. 

The normalization score formula still confuses me. Why do you add S/Si to calculate? Is it used in other similar papers?

Fig.1a could be replaced with a venn diagram

7K8N and 7D0C is pdb number, not antibody name. Please use their name instead. 

Usually for CDR3 sequence we don't include the first "C" and the last "W" as written in this paper. 

Comments on the Quality of English Language

Minor adjustments needed (as attached). There are some grammatical mistakes. 

Reviewer 3 Report

Comments and Suggestions for Authors

A very carefully prepared manuscript by scientists from the USA. I don't have any major comments about the methodology or the results - they are described in quite a detail.

Nevertheless, in the discussion it is worth adding the limitations of the study and the last paragraph with the conclusion, also how these results can translate into the everyday clinic of patients.
